# Health in Preconception, Pregnancy and Postpartum Global Alliance: International Network Pregnancy Priorities for the Prevention of Maternal Obesity and Related Pregnancy and Long-Term Complications

**DOI:** 10.3390/jcm9030822

**Published:** 2020-03-18

**Authors:** Briony Hill, Helen Skouteris, Jacqueline A. Boyle, Cate Bailey, Ruth Walker, Shakila Thangaratinam, Hildrun Sundseth, Judith Stephenson, Eric Steegers, Leanne M. Redman, Cynthia Montanaro, Siew Lim, Laura Jorgensen, Brian Jack, Ana Luiza Vilela Borges, Heidi J. Bergmeier, Jo-Anna B. Baxter, Cheryce L. Harrison, Helena J. Teede

**Affiliations:** 1Monash Centre for Health Research and Implementation, School of Public Health and Preventive Medicine, Monash University, Level 1, 43-51 Kanooka Grove, Clayton, VIC 3168, Australia; briony.hill@monash.edu (B.H.); helen.skouteris@monash.edu (H.S.); jacqueline.boyle@monash.edu (J.A.B.); cate.bailey@monash.edu (C.B.); ruth.walker@monash.edu (R.W.); siew.lim1@monash.edu (S.L.); heidi.bergmeier@monash.edu (H.J.B.); cheryce.harrison@monash.edu (C.L.H.); 2Warwick Business School, Warwick University; Coventry CV47AL, UK; 3Barts Research Centre for Women’s Health (BARC), Women’s Health Research Unit, Centre for Primary Care and Public Health, Blizard Institute, Barts and The London School of Medicine and Dentistry, 58 Turner Street, London E1 2AB, UK; S.Thangaratinam.1@bham.ac.uk (S.T.); laurajjorgensen@gmail.com (L.J.); 4European Institute of Women’s Health, 33 Pearse Street, Dublin 2, Ireland; h.sundseth@gmx.de; 5Institute for Women’s Health, University College London, EGA Institute for Women’s Health, 74 Huntley St, London WC1E 6AU, UK; judith.stephenson@ucl.ac.uk; 6Department of Obstetrics and Gynaecology, Erasmus Medical Centre–Sophia Children’s Hospital, Wytemaweg 80, 3015 CN Rotterdam, The Netherlands; e.a.p.steegers@erasmusmc.nl; 7Reproductive Endocrinology and Women’s Health Laboratory, Pennington Biomedical Research Center, 6400 Perkins Rd, Baton Rouge, LA 70808, USA; leanne.redman@pbrc.edu; 8Wellington-Dufferin-Guelph Public Health, 160 Chancellors Way, Guelph, ON N1G 0E1, Canada; cynthia.montanaro@wdgpublichealth.ca; 9Department of Family Medicine, Boston University School of Medicine, 771 Albany St, Boston, MA 02118, USA; bjack@bu.edu; 10Public Health Nursing Department, University of Sao Paulo, 419 Cerqueira Cesar, Sao Paulo 05403000, Brazil; alvilela@usp.br; 11Centre for Global Child Health, The Hospital for Sick Children, Peter Gilgan Centre for Research and Learning, 686 Bay Street, Toronto, ON MG5 0A4, Canada; jo-anna.baxter@sickkids.ca; 12Department of Nutritional Sciences, University of Toronto, Medical Sciences Building, 1 King’s College Circle, Toronto, ON M5S 1A8, Canada; 13Monash Partners Advanced Health Research Translation Centre, Locked Bag 29, Clayton, VIC 3168, Australia; 14Monash Health, 246 Clayton Road, Clayton, VIC 3168, Australia

**Keywords:** pregnancy, antenatal care, obesity prevention, lifestyle behaviours, consensus, research priorities

## Abstract

In this article, we describe the process of establishing agreed international pregnancy research priorities to address the global issues of unhealthy lifestyles and rising maternal obesity. We focus specifically on the prevention of maternal obesity to improve related clinical pregnancy and long-term complications. A team of multidisciplinary, international experts in preconception and pregnancy health, including consumers, were invited to form the Health in Preconception, Pregnancy and Postpartum (HiPPP) Global Alliance. As an initial activity, a priority setting process was completed to generate pregnancy research priorities in this field. Research, practice and policy gaps were identified and enhanced through expert and consumer consultation, followed by a modified Delphi process and Nominal Group Technique, including an international workshop. Research priorities identified included optimising: (1) healthy diet and nutrition; (2) gestational weight management; (3) screening for and managing pregnancy complications and pre-existing conditions; (4) physical activity; (5) mental health; and (6) postpartum (including intrapartum) care. Given extensive past research in many of these areas, research priorities here recognised the need to advance pregnancy research towards pragmatic implementation research. This work has set the agenda for large-scale, collaborative, multidisciplinary, implementation research to address the major public health and clinical issue of maternal obesity prevention.

## 1. Introduction

In developed nations, approximately 50% of women enter pregnancy above the healthy recommended weight [1,2,3]. Obesity is also a major health burden in low- and middle-income countries, coexisting alongside undernutrition [4]. Pregnancy is a key driver of obesity in women [5,6]. Factors such as high preconception body mass index (BMI), excessive gestational weight gain and postpartum weight retention contribute independently and significantly to adverse maternal outcomes [7,8] and are also associated with adverse health outcomes in offspring [9,10]. Consequently, prevention of maternal obesity is a critical strategy to improve health outcomes for women and their children.

International and national groups such as the World Health Organization (WHO) [11,12], the UK National Institute for Health and Care Excellence [13], the US Institute of Medicine (now National Academy of Medicine) [2], Health Canada [14,15] and the Australian National Health and Medical Research Council [16] have identified preconception and pregnancy as key opportunities for obesity prevention. However, there is limited international consistency regarding approaches to address lifestyle factors and obesity during this life phase [7]. Whilst the relationship between high gestational weight gain and adverse maternal and infant outcomes is well understood and associated benefits of lifestyle intervention to optimise gestational weight gain are positive [17], translation and implementation for broad benefit remains inadequate. For example, clinical practice guidelines for weight management in pregnancy do not guide implementation of the evidence on health benefits of lifestyle modification in pregnancy into practice [13,14]. Identification of key pregnancy research priorities for the prevention of maternal obesity and related pregnancy and long-term complications is important to advance the field and generate international guidelines and policy directives that will inform practice and deliver public health impact.

To meet this need, the Health in Preconception, Pregnancy and Postpartum (HiPPP) Global Alliance was formed [18,19] and an international forum was convened in Prato, Italy in September 2018. This inaugural meeting of the HiPPP Global Alliance, expanded from the Australian national HiPPP Collaborative established in 2013 [19], included key experts in preconception, pregnancy and postpartum lifestyle, health and obesity prevention, from diverse disciplines worldwide. The five initial objectives of the Alliance were to: (i) develop a set of agreed priorities for research in preconception and pregnancy lifestyle, health, and care with the ultimate goal to prevent maternal obesity and related short- and long-term complications; (ii) review guidelines internationally on lifestyle modification in the preconception period and during pregnancy to grade quality and identify gaps; (iii) develop an agreed consumer involvement and advocacy strategy for HiPPP research and translation activities; (iv) develop agreed workforce capacity building strategies; and (v) develop capacity in early career researchers (ECRs) in HiPPP fields.

The objective of this paper is to describe the activities relating to the first goal, focused on pregnancy. That is, we describe the process of establishing agreed international research priorities for pregnancy to target the global challenge of unhealthy lifestyles and rising maternal obesity, specifically for the prevention of maternal obesity and related clinical pregnancy outcomes and long-term complications. Priorities for preconception research are presented in an accompanying paper in this issue [20].

## 2. Method

### 2.1. Research Priority Setting Process

The research priority setting process has been applied elsewhere [21,22,23,24] and is presented below [20]. Thirteen key stakeholders of international standing in preconception and pregnancy health and six early career researchers with relevant expertise were invited to join the HiPPP Global Alliance; members of the Alliance participated in the research priority setting exercise. A modified Delphi process and Nominal Group Technique were used to determine the pregnancy priorities [21,22]. The process (Figure 1) involved literature, policy, expert and consumer identification of initial priorities for evaluation, with a pre-workshop electronic Delphi ranking process. An international workshop involved discussion, sense-making and additional surveys for ranking the priorities. Post-workshop consultation via electronic communication finalised details and developed this publication on research priorities.

#### 2.1.1. Inputs

Thirteen expert who were world leaders in their field based on publications, considering geographic and discipline diversity, were invited to take part. Two consumer representatives from non-governmental women’s health and consumer representative organisations, who were trained in participating as consumer experts in research activities were also invited. Six early career researchers with relevant expertise were also invited; all early career researchers were from the Monash Centre for Health Research and Implementation and provided only one combined vote. Their primary role was support during the workshop. All workshop participants have been included as authors on this manuscript. The only exception is our expert from Africa who represented WHO in the priority setting process but is not an author on the manuscript as he had moved on from the position. The initial list of priorities for consideration was derived from a comprehensive systematic review of existing preconception and pregnancy guidelines, and WHO recommendations on preconception and pregnancy care [12,25].

#### 2.1.2. Pre-Workshop Ranking (Round 1)

One month prior to the workshop, experts were sent a list of eight pregnancy research priorities determined from the inputs above via email. Using a modified Delphi format, experts were asked to privately rank the priorities according to their opinion on relevance for maternal obesity prevention from one (highest ranking) to eight. Early career participant input was weighted to be equivalent to one international expert across ECRs combined. Experts were also asked to suggest additional priorities for consideration. Mean scores for the Round 1 ranking were calculated (lower score equated to higher ranking).

#### 2.1.3. Workshop Processes

##### First Group Discussion, Vote and Sense-Making (Round 2)

At the workshop, experts were provided with the pre-workshop rankings and divided into two groups to discuss the research priorities. Using a Nominal Group Technique process, groups reviewed the priorities and completed a sense-making activity to reach shared definitions and understanding for priorities and to consider whether any could be consolidated [21]. A facilitated whole group discussion followed to consolidate and integrate inputs with all members’ voices captured. During this first group discussion, it was also agreed that an overarching set of principles was required to be considered against all research priorities.

##### Second Group Discussion and Vote (Round 3)

A Policy Prioritisation Framework was adapted from a policy prioritisation process applied in Australian national priority setting in women’s health [23,24]. This framework was used to draw participants’ attention to evidence-translation gaps when establishing the research priorities. It offered nine criteria for priority assessment: prevalence, prevention, position, provision, potential, participation, policy, proposed strategy and proposed transformation attributable to the problem. The criteria (9Ps) have been adapted, refined and applied by the national Australian Health Research Alliance for priority setting in women’s health [24]. The criteria were endorsed by HiPPP members. Experts were provided with the Round 2 rankings and then discussed in groups the priorities with reference to the 9Ps priority setting framework. Then, experts completed a final (Round 3) private ranking of the research priorities. Mean ranking scores were computed.

##### Consensus Development of Research Priorities

Facilitated discussion resulted in minor modifications to the priorities, with a majority vote to approve proposed changes. Experts were asked to form a consensus on the top priorities that would be the focus of strategic, prioritised future research. The final workshop group discussion focused on identifying specific research priority actions, including research relevant to practice and policy across the pregnancy research priorities identified through the ranking process.

## 3. Results

### 3.1. Research Priority Setting Process

In total, there were 20 participants in the workshop (of the 21 invited, one (from Asia) was unable to attend at the last minute). Experts who contributed to the priority setting process included individuals from diverse disciplines, including endocrinology, psychology, obstetrics, gynaecology, exercise physiology, dietetics, health economics, epidemiology, nursing, public health, as well as two experts in consumer advocacy and consumer experience. Disciplines were approximately equally represented. The two consumer experts (also referred to as consumer and community involvement (CCI) or patient and public involvement (PPI)) were from established non-governmental consumer representative and women’s health organisations. As part of their involvement with these organisations they had received training on participation as consumer experts in research activities. Experts were from five continents, with countries including Australia, Belgium, Brazil, Canada, The Netherlands, South Africa, United Kingdom and United States of America. Five ECRs with expertise in preconception and pregnancy health were also invited.

#### 3.1.1. Inputs

The initial eight pregnancy research priorities (before ranking and workshop) for maternal obesity prevention were (in no particular order): dietary interventions, physical activity, gestational weight management, gestational diabetes mellitus and diabetes, mental health, nutritional supplementation and anaemia, sexually transmitted infections and blood borne viruses, tobacco and substance use.

#### 3.1.2. Pre-Workshop Ranking (Round 1)

Mean ranking scores for the pre-workshop ranking and are presented in Table 1. Additional pregnancy research priorities for consideration suggested by the experts were ‘reproductive and obstetric history’ and ‘medications and vaccinations’.

#### 3.1.3. Workshop Processes

##### First Group Discussion, Vote and Sense-Making (Round 2)

During the first group discussion and sense-making activity, a priority called ‘healthy diet and nutrition’, which also includes supplementation, emerged by amalgamating ‘dietary interventions’ and ‘nutritional supplementation and anaemia’ priorities. A priority called ‘screening and management of pregnancy complications and pre-existing conditions’ was also created which encompasses, but is not limited to, gestational diabetes, hypertension, and fetal growth monitoring, risk profiling and use of medications. Furthermore, a priority on ‘postpartum (including intrapartum) care’ was added that includes, but is not limited to, breastfeeding, postnatal depression and sleep. Round 2 rankings are presented in Table 1. During workshop discussions, four overarching principles were also determined. These were: (a) context of broader preconception and pregnancy priorities; (b) social determinants of health; (c) health of families; and (d) cultural considerations [20].

##### Second Group Discussion and Vote (Round 3)

After considering the priorities against the Policy Prioritisation Framework, the Round 3 ranking process generated the eight research priorities presented in Table 1.

##### Consensus Development of Research Priorities

The group decided that the top six priorities would form the final list (Box 1). The final workshop group discussion identified specific research, practice and policy priorities. After the workshop, these were circulated to the expert group for final consultation with no substantial changes made (see Box 2).

Box 1Final consensus for pregnancy research priorities.
Promoting healthy diet and nutrition, includingSupplementationOptimising gestational weight gain managementScreening for pregnancy complications and pre-existing conditions, includingGestational diabetes mellitus and diabetes, hypertension, fetal growth monitoringRisk profiling (e.g., deep vein thrombosis, sleep apnoea)MedicationsOptimising physical activityOptimising mental healthPostpartum (including intrapartum) care, includingBreastfeeding supportPostnatal depression screening and managementOptimising sleep


Box 2Research gaps to be targeted to help address the identified priorities.
Primary studies are needed to improve basic understanding of mechanisms, pathways, biological drivers, and impacts on outcomes, including offspring outcomes. This includes exploring dietary components that contribute specifically to weight gain during pregnancy and the physiological mechanisms associated with diet and weight gain in pregnancy.Factors that impact lifestyle change throughout pregnancy, such as social support, and mental and physical health, as well as taking a life course approach (including transitions from preconception and to postpartum) need further understanding of how they can be applied or addressed in lifestyle interventions in pregnancy.Synthesis is required for all levels of evidence including observational studies, randomised controlled trials and pragmatic implementation trials. Evidence synthesis of intervention studies are imperative, rather than additional randomised controlled trials. Here, secondary research should be prioritised to understand the effective components of lifestyle intervention in pregnancy and establish cost-effectiveness.Implementation research is vital to drive evidence of efficacy into broader effectiveness and practice. This will include incorporating systems level approaches, health professional training, and adaptation of policies. A biopsychosocial lens must be applied amid multidisciplinary, population approaches to target the issues at hand.Consumer engagement is essential early in the research process and across the implementation and scale up pathway. Consumer engagement refers to an active partnership between the researchers and the individuals affected or impacted by the research (e.g., recipients of a health service), rather than the traditional mode of having research conducted to or for them [26]. It may take many forms during all stages of the research cycle, from research participant to partner in research design [27].Co-design techniques are paramount to the design of interventions that are based on stakeholder partnership, and that can be implemented in a cost-effective manner at scale.Once the body of evidence is sufficient, risk or score cards for pregnancy lifestyle health can be developed and used to cost-effectively and efficiently triage individuals for appropriate screening, treatment or models of care to improve maternal, fetal and long-term outcomes at an individual and public health level.


## 4. Discussion

International research priorities in pregnancy, for the prevention of maternal obesity and related pregnancy and long-term complications were identified by the multidisciplinary HiPPP Global Alliance experts through a multistep, transparent, modified Delphi and Nominal Group Technique consensus development process. Six research priorities were identified. Key overarching principles across the priorities included the context of broader preconception and antenatal care priorities, social determinants of health, the health of families, and cultural considerations, further discussed in our preconception priority setting paper in this journal special issue [20].

The highest ranked pregnancy research priority was promoting healthy diet and nutrition, which includes supplementation. Globally, women’s dietary intake is typically sub-optimal during pregnancy, despite the additional nutritional needs associated with gestation [28,29]. In low- and middle-income countries, nutritional deficiencies often co-exist with obesity [4]. Suboptimal dietary intake with excessive energy intake during pregnancy is a key driver of maternal obesity, and extensive randomised trials have shown that lifestyle interventions in pregnancy are effective in reducing excess gestational weight gain and improving health outcomes [17]. Secondary research is now needed to progress understanding on the most effective components in these interventions (e.g., behaviour change strategies) that promote dietary and nutritional improvement during pregnancy. Implementation research is vital, including cost effectiveness studies to move this evidence into practice. Greater understanding of how to deliver culturally relevant interventions that sit within the scope and resources of diverse settings is needed [12]. Public health approaches to diet and nutrition that encompass the broader population (e.g., tax on sugar sweetened beverages) are recognised as vital to benefit pregnant populations, with greater policy and implementation research needed.

Gestational weight management was ranked the second research priority. Consensus at the workshop indicated that gestational weight gain should be a separate priority given the independent effects of excessive pregnancy weight gain on maternal and fetal outcomes [7]. Excessive gestational weight gain is a key contributor to increased weight retention postpartum, fueling the progressive increase in weight observed in women of reproductive age [6,30]. A recent individual participant data meta-analysis of 36 randomised controlled trials demonstrated a reduction in gestational weight gain utilising intervention approaches for behaviour change including diet, physical activity and behaviour change strategies [17]. As with dietary interventions, unpacking the “active ingredients” for lifestyle intervention focusing on individual behaviour change in pregnancy is now required. Implementation research targeting health care professionals is also important. For instance, training and support is urgently required to assist care providers (e.g., midwives, obstetricians, general practitioners) to engage in healthy lifestyle conversations with pregnant women and reinforce messages around healthy gestational weight gain [31]. Additionally, systems approaches and guidelines that support and enable these changes to practice are required, including inter-professional collaboration and referral pathways [32].

Screening for pregnancy complications and pre-existing conditions was ranked third. This research priority includes specific attention to gestational diabetes mellitus, hypertension, fetal growth monitoring and other complications, all with direct implications on pregnancy and offspring outcomes and that interact with maternal lifestyle, BMI and gestational weight gain [7]. Risk profiling and medications (including use of antipsychotic and antidepressant medications that are linked with weight gain [33]) are also included here. This priority focuses on ensuring pregnancy-specific factors relating to health are adequately addressed.

Optimising physical activity was ranked the fourth research priority. Prior meta-analyses have indicated the importance of physical activity in pregnancy in optimising gestational weight gain and improving birth outcomes [17]. While often considered in the whole context of lifestyle, additional research should focus specifically on physical activity in pregnancy, given the decline that commonly occurs in women during this life phase [34,35,36]. Recent research suggests that declining activity in pregnancy now negates any need to increase energy intake, particularly in the first trimester [37]. In particular, interventions that encourage women to maintain adequate antenatal physical activity levels and that are not dependent on delivery by health care professionals are needed for feasible implementation and scale up. It is recognised that broader population strategies to increase physical activity are also vital for this population.

Optimising mental health was ranked the fifth research priority. Mental health concerns, including symptoms of depression and anxiety, affect up to one-quarter of pregnant women [38,39], with up to 10% of women experiencing other co-morbid mental health conditions [40]. Women who experience mental health concerns during pregnancy are at greater risk of postnatal depression and associated adverse outcomes for themselves and their infants [41,42]. Diet quality in pregnancy is associated with poorer mental health, with both causal [43] and bi-directional [44] relationships implicated. There is also evidence for the use of physical activity to ameliorate depression [45,46]. Research is needed on improving lifestyle to address mental health concerns during pregnancy, and to improve mental health to positively impact lifestyle and weight outcomes during this life phase. Stigma associated with preexisting excess weight also affects mental health and needs greater exploration, including optimising women’s experiences in this area and strategies for health professionals and women to focus on positive healthy lifestyle and healthy weight gain and avoid shame and stigma associated with excess weight [47,48]. Future research should also explore interactions and evaluate relevant interventions that encompass healthy lifestyle and physical and mental health to inform guideline, practice and policy.

Postpartum (including intrapartum) care was ranked as the sixth research priority. This priority includes but is not limited to important factors such as clinical management during labour and delivery, breastfeeding support, postnatal depression screening and care, postnatal contraception and sleep hygiene. The intrapartum and postpartum periods impact on the health of women and their children, which may directly or indirectly affect lifestyle behaviours and weight status [49]. For example, intrapartum synthetic oxytocin dosage is positively associated with reduced exclusive breastfeeding and adverse maternal psychosocial well-being at two months postpartum [50]. Furthermore, the postpartum period is a key inter-conception phase with implications for preconception health and care [51]. While the areas of breastfeeding, postnatal depression and sleep are well studied on their own [52,53,54], further investigation into how we can address and implement these and other modifiable factors to promote healthy lifestyles, and therefore optimise weight status, for women between and in future pregnancies is warranted. As with antenatal interventions, understanding the key effective components of interventions to reduce postpartum weight retention, alongside vital implementation research focused on overcoming barriers to postpartum self-care and healthy lifestyle, are vital [55]. Furthermore, continuity of care from pregnancy through intrapartum and postpartum should be optimised [56,57,58].

Key research gaps, including those relevant to practice and policy were discussed by the global HiPPP Alliance. Key gaps were evident across the research-implementation cycle and included some need for primary research and co-designed interventions, with a clear mandate for secondary research and evidence synthesis to capitalise on existing evidence. Across all priorities, implementation research was recognised as vital (Box 2). Notably, framework and component analysis of existing intervention studies to complement meta-analyses (e.g., [7,17]) are arguably preferable over conduct of additional randomised controlled trials in pregnancy lifestyle interventions. This secondary research should focus on understanding the effective components of antenatal lifestyle intervention. Implementation research including cost-effectiveness is now the greatest priority in this area with 117 randomised controlled trials completed to date. The HiPPP Global Alliance is committed to actively seeking collaborative opportunities both within and outside the Alliance in these priority areas to capitalise on knowledge, expertise, resources and funding opportunities, and to publish, share and implement knowledge gains across academic, policy and health sectors [18]. This work complements Alliance preconception research priorities with both these life stages of importance for women and their children [20]. This research needs to be undertaken in a range of settings globally, including low-middle and high-income countries, as well as regional, rural and urban environments. Importantly, outputs arising from prioritised research will inform key areas for implementation by care teams, hospitals and public health authorities in the future.

The HiPPP Global Alliance members brought their breadth of expertise and consumer experience in pregnancy, lifestyle and weight to the rigorous consensus development process and committed to sharing their knowledge and fostering collaborations rich in interdisciplinary, cross-cultural, and international perspectives. Many of the pregnancy research priorities can be addressed promptly, with work underway including secondary research on the 117 randomised controlled trials on lifestyle intervention in pregnancy via component analysis (including behaviour change techniques, intervention components and implementation characteristics), framework review and cost-effectiveness analysis. This evidence will assist with determining the optimal pregnancy lifestyle intervention core components and those that are adaptable during implementation, feeding into implementation research to deliver programs ready for scale up. Additional ongoing research includes evaluating health-related outcomes other than weight following lifestyle intervention during pregnancy as well as understanding antenatal energy balance requirements and development of accurate, practical recommendations for caloric intake. Alliance members committed to the agreed overarching principles including engaging women of diverse cultural backgrounds in their research across the preconception, pregnancy and postpartum periods [59].

### Strengths and Limitations

The research priorities were generated through a robust process that took into consideration existing international and national guidelines, WHO recommendations, and diverse expert and consumer input. The evidence-based consensus development technique minimised bias by managing undue dominance of individuals or groups through facilitated discussion and an equal confidential vote from each participant in each Delphi round [22]. Consumers and diverse health disciplines and cultures were included. Limitations were that all relevant disciplines may not have been represented equally at the workshop, not all invited participants took part in the process, and some invited participants were unable to attend. Furthermore, other international experts may not have been invited who may have brought additional contributions and insight to the process, however it was important to balance global input and prevent domination of some geographic regions with a higher proportion of experts.

## 5. Conclusions

An international research priority setting process, including a workshop, was convened and the HiPPP Global Alliance formed to address healthy lifestyle and the prevention of maternal obesity and related adverse short- and long-term outcomes. A multistage Delphi survey and Nominal Group Technique process was used to identify the top six pregnancy research priorities and principles to collaboratively advance knowledge and translation in the field. Priorities include promoting healthy diet and nutrition; gestational weight management; screening for pregnancy complications and pre-existing conditions; optimising physical activity; optimising mental health; and postpartum (including intrapartum) care. Given the broad global reach of unhealthy lifestyle, high body mass index and excessive gestational weight gain, these pregnancy research priorities are relevant across high and low-middle income countries [1,2,3,4]. Despite current work by the Alliance and others focused on these priorities, key gaps remain, including the need for secondary and implementation research such as understanding the effective components of antenatal lifestyle intervention and applying co-design techniques for adequate implementation, to drive current research into practice. To address these priorities, members of the HiPPP Global Alliance have committed to collaborative, prioritised research, implementation and scale up to improve health and clinical outcomes for women and their children.

## Figures and Tables

**Figure 1 jcm-09-00822-f001:**
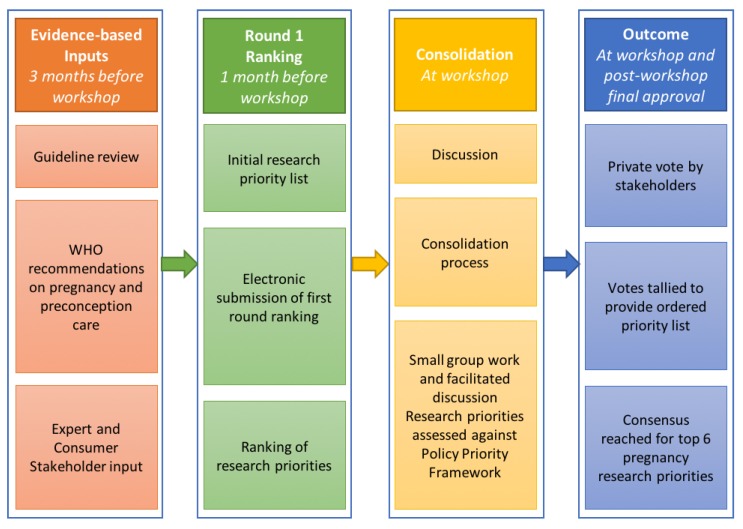
Process of establishing consensus on pregnancy research priorities.

**Table 1 jcm-09-00822-t001:** Round 1, 2 and 3 pregnancy research priority rankings.

Pregnancy Research Priority	Round 1 Ranking	Round 2 Ranking	Round 3 Ranking
Promoting healthy diet and nutrition Supplementation	1 ^a^, 5 ^b^	1	1
Optimising gestational weight management	2	2	2
Screening for pregnancy complications and pre-existing medical conditions Including gestational diabetes mellitus and diabetes, hypertension, fetal growth monitoring Risk profiling (e.g., deep vein thrombosis, sleep apnoea) Medications	3 ^c^	3	3
Optimising physical activity	4	4	4
Optimising mental health	6	5	5
Postpartum (including intrapartum) care Breastfeeding Postnatal depression Optimising sleep	*Not ranked* ^d^	6	6
Substance use (including alcohol and tobacco)	7	7	7
Infections Sexually transmitted infections and blood-borne viruses	8	8	8

^a^ Ranked as ‘dietary interventions’ in round 1. ^b^ Ranked as ‘nutritional supplementation and anaemia’ in round 1. ^c^ Ranked as ‘gestational diabetes mellitus and diabetes’ in round 1. ^d^ This priority was developed as part of the sense-making activity and hence was not ranked in round 1.

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
