# Peer review of "Health in Preconception, Pregnancy and Postpartum Global Alliance: International Network Pregnancy Priorities for the Prevention of Maternal Obesity and Related Pregnancy and Long-Term Complications"

_jcm, 2020, doi:10.3390/jcm9030822_

Round 1
Reviewer 1 Report
I agree with the changes made to the manuscript.
Thank you for the thorough responses.
Reviewer 2 Report
Thank-you for the opportunity to review this manuscript. I read with interest your approach to achieving consensus in listing priority areas in research related to the prevention of pregnancy-associated weight gain and obesity. Though I believe this process had the potential to uncover some very important findings, the results and related discussion presented here offer very non-specific guidance with little emphasis on key actionable items for readers to take-away. This is most notable in Box 2 where very broad language is used in each point that could apply to almost any field in medicine – none of these points are specific to maternal health which makes me question what this manuscript truly adds to the current literature.
I believe significant revisions need be made in the results and discussion sections to highlight the importance of this consensus achieving process and how it adds to the current literature by emphasizing key actionable items that care teams/hospitals/public health authorities can take-away after reading this statement would be most welcomed.
Do these findings/priorities apply to low- and middle-income countries as well? Or it is targeted at high-income countries. This is not clear and should be made explicit. The decision here should also be guided by the fact that it appears most/all authors are from high-income countries.
I also have further comments on a few key items:
- How were the 13 key stakeholders and 6 early career researcher chosen? This process is not currently described in the methods. An explanation would help add to the transparency of the methodology. And is a list of all experts and ECRs available? That should also be included.
- Also there is no mention in the methods that consumer experts would be chosen and how they would be chosen. This only comes up in the results section. Please add this to the methods to clarify and improve transparency.
- Provided a timeline alongside Figure 1 (ie. adding in dates above each box/phase) would be very helpful.
- In line 187, do you mean Box 1 and not Table 1? I do not see a Table 1 in the current draft, just Box 1 and 2.
- In Box 2, I would suggest that point 2 is very vague and broad. Everything from a life course approach and social support to transitioning into postpartum care is mentioned. Refining the wording here to improve clarity and practicality would be of benefit - as noted above.
Round 2
Reviewer 2 Report
Thank-you very much for taking the time to respond to each point in turn and to make these thoughtful revisions where necessary. The manuscript is much stronger with improved overall clarity.
A couple minor points to mention:
The emphasis that these are research priorities should be made clear throughout the entire manuscript to ensure clarity to readers. For example, Table 1 could be entitled "Round 1, 2 and 3 research priority rankings" and the first column could be entitled "Pregnancy Research Priority" instead of "Pregnancy Priority". Box 1 could be entitled "Final consensus for pregnancy research priorities" with the box heading in bold "Pregnancy Research Priorities" (or this bolded heading inside the box could be eliminated completely as it is redundant with the title).
A reference for the statement made in lines 355-357 would be helpful. Are unhealthy lifestyle, high BMI and excessive gestational weight gain globally ubiquitous? I like this statement that you've added but wonder if it might be strengthened by a reference since "ubiquitous" is a strong word. Alternatively you could soften the wording a bit.
Author Response
Please see the attachment.

This manuscript is a resubmission of an earlier submission. The following is a list of the peer review reports and author responses from that submission.
Round 1
Reviewer 1 Report
This paper describes research priorities for maternel health from preconception to postpartum.
The paper describes the process of selection of research priorities during a maternel health conference.
The process is well described, however I do not see the need for such a paper. A review describing the existing knowledge and new research needs withins each research priority would have been interesting, but reading about the selection of research priorities seems unrelevant.
Reviewer 2 Report
The article “Health in Preconception, Pregnancy and Postpartum (HiPPP) Global Alliance: International network pregnancy priorities for the prevention of maternal obesity and related pregnancy and long-term complications” presented by Hill et al. is a priority setting exercise. This article describes the methods used to set these priorities and then provides a summary of the final consensus. Throughout the article I would encourage the authors to think about what they want the main take-away to be for researchers reading this article. I like the use of the boxes to help focus the paper results and discussion. As a researcher, I found Box 2 to be the most useful. I applaud the authors on their work on this very important topic. I have some minor comments and suggestions to help improve the paper.
Line 64: I find the first sentence awkward – do you present “with overweight and obesity” or “as overweight or obese”?
Line 75: Suggest word changing: However, there is limited international consistency regarding approaches to address lifestyle factors and obesity during this life phase.
Line 79. Can you provide a reference to reinforce your statement that translation and implementation have been slow.
Line 91: lifestyle modification in the preconception period and …
Line 92-93: I find this sentence very wordy. Can you modify (suggestion follows): Develop a consumer involvement and advocacy strategy for HiPPP research and knowledge translation.
Line 94-96: I think the objective deserves its own paragraph. I found myself getting lost within all the words. I think it needs to be very clear that they objectives of this paper are focused on research priorities to prevent maternal obesity during pregnancy (correct?). Perhaps that could be simply stated here.
The objective of this paper describe the activities related to first objective but focus on research priorities to prevent maternal obesity during pregnancy. We describe the process….
Line 108: ‘additional ranking of surveys’.
Results: Line 163: it would be nice to know the number of people involved in the process (unless I missed that). Although they were from diverse backgrounds were they mainly dominated in one field or was there an even spread of experts across disciplines?
I don’t feel that I’m clear on who the consumer is. When you say ‘consumer experience’ are you referring to the patient (pregnant women) or to health care professionals? Who were the consumer advocacy and consumer experience groups – please expand on this so other researchers will understand the process better.
In your Box 1 and throughout the article when you are referring to the priories I think it would be better received if were more specific and included the ‘action’ words -
Promoting Healthy diet and nutrition Optimising Physical Activity Optimising Mental Health Breastfeeding Support (?) Postnatal depression Screening (?) Optimising Sleep
In Box 2 – again what do you mean by consumer engagement (be more specific or provide a reference/guide)
Line 290. Missing period after …HIPPP Alliance. Key gaps
Line 322: Expand on the bias that may have been introduced and why this technique minimised it.
In your limitations you mention that other international experts may not have been invited – what other methods could have been used to involve world experts? What other steps could have been taken to include them – in a future workshop? I think this information would make the paper more useful to other researchers who may need to plan a similar workshop on different topics.
Conclusion: You may want to re-iterate your preference for secondary research and meta-analyses for effective antenatal lifestyle interventions.